# PINNverse: Accurate Parameter Estimation in Differential Equations from Noisy Data with Constrained Physics-Informed Neural Networks

## Abstract

Estimating unknown parameters in differential equations from noisy, sparse data is a common inverse problem in science and engineering. While Physics-Informed Neural Networks (PINNs) have shown promise, their standard training paradigm, which relies on a weighted-sum loss, often leads to overfitting and fails to enforce physical laws in the presence of noise. This failure stems from the inability of gradient-based methods to find balanced solutions on the complex, often non-convex, Pareto fronts that arise in such multi-objective settings. We introduce PINNverse, a new training paradigm that overcomes these limitations by reformulating the learning process as a constrained optimization problem. Instead of balancing competing objectives with ad-hoc weights, PINNverse minimizes the data-fitting error subject to the explicit constraint that the differential equations and boundary conditions are satisfied. To solve this, we employ the Modified Differential Method of Multipliers (MDMM). By simultaneously updating network weights and Lagrange multipliers (via gradient ascent) in a single optimization loop, this method avoids the expensive nested loops required by conventional augmented Lagrangian techniques and seamlessly integrates with standard optimizers like Adam. This enables convergence to any point on the Pareto front—including concave regions inaccessible to standard PINNs—while adding negligible computational overhead. Experiments on four challenging ODE and PDE benchmarks demonstrate that PINNverse achieves robust and accurate parameter estimation even with significant data noise and poor initial guesses, successfully preventing overfitting and ensuring strict adherence to the governing physics. By solving the forward and inverse problems concurrently, PINNverse enables efficient parameter inference in systems where repeated forward evaluations with classical numerical solvers would be computationally prohibitive.

## 1 Introduction

Solving inverse problems for differential equations (DEs) is a cornerstone of scientific discovery and engineering, allowing for the inference of physical parameters from observational data. While classical frequentist (Fisher, 1922; Aldrich, 1997) and Bayesian (Jeffreys, 1939) methods have been foundational, they are often hampered by challenges such as non-convex optimization landscapes (Tu & Mayne, 2002; Villaverde et al., 2019) and significant computational demands (Stuart, 2010; Luengo et al., 2020).

Recently, Physics-Informed Neural Networks (PINNs) (Raissi et al., 2019; Karniadakis et al., 2021) have emerged as a powerful, mesh-free paradigm for solving DEs by embedding the governing physical laws directly into the neural network's loss function. However, when applied to inverse problems with noisy data, a fundamental tension arises between fitting the observations (data loss) and satisfying the physical constraints (physics loss). Simply balancing these terms often leads to overfitting the noise at the expense of physical accuracy. For many physical systems, the governing DEs are non-negotiable constraints, not objectives to be traded off for. We therefore reframe the PINN inverse problem as a constrained optimization task: minimize the data loss subject to strict

compliance with the physical laws. To solve this, we introduce the Modified Differential Method of Multipliers (MDMM) (Platt & Barr, 1987) as a novel training strategy for PINNs. MDMM elegantly integrates the constraints using Lagrange multipliers that are updated in parallel with the network and DE parameters. This approach offers four key advantages:

1. It avoids the nested optimization loops of traditional augmented Lagrangian methods (Lu et al., 2021), imposing little computational overhead compared to standard PINN training as we will see.

2. It is fully compatible with state-of-the-art, first-order optimizers like Adam (Kingma & Ba, 2014).

3. It robustly infers parameters even with substantial data noise and poor initial guesses, scenarios where even robust classical numerical optimizers fail.

4. It provides a straightforward mechanism to enforce practical parameter bounds, a common requirement in real-world inverse problems.

We demonstrate on four benchmark inverse problems that MDMM-trained PINNs substantially outperform standard PINNs and classical optimizers, establishing a more robust and physically principled framework for parameter inference.

## 2 RELATED WORK

**Challenges and Enhancements in PINN Training.** Despite broad adoption, standard (soft-constraint) PINN training can fail due to ill-conditioned gradients and stiff loss landscapes; NTK analyses and empirical diagnostics make these pathologies explicit (Wang et al., 2022; Krishnapriyan et al., 2021). Remedies include causality-aware training for time-dependent systems (Wang et al., 2024), high-frequency feature embeddings (Fourier features/positional encodings) (Tancik et al., 2020), adaptive resampling of collocation points (e.g., R3 sampling) (Daw et al., 2023), and numerous adaptive loss-balancing schemes (D. & Braga-Neto, 2023; Xiang et al., 2022; Bischof & Kraus, 2025). While effective in practice for forward solves, these techniques retain a penalty-based, weighted-sum objective that reshapes representation/sampling or rescales terms without an explicit noise model; consequently, in inverse settings with noisy observations the data and physics losses are generically incompatible in the zero-residual limit and equal weighting promotes noise-fitting. Moreover, heuristic or static weighting explores only a fixed weighted-sum compromise and lacks statistically calibrated data losses or strict physics satisfaction, yielding biased parameter estimates and persistent residual violations under noise.

**Multi-objective formulations and Pareto analysis.** To make the data–physics tension explicit, several works adopt a multi-objective viewpoint. Rohrhofer et al. map the apparent Pareto front induced by system scalings and loss weights (Rohrhofer et al., 2021), Heldmann et al. treat data and residual losses as truly biobjective (Heldmann et al., 2023). Evolutionary strategies approximate or explore the front directly, e.g., NSGA-PINN and related Pareto-inspired algorithms (Lu et al., 2023; Lazovskaya et al., 2023), with a recent survey reviewing neuro-evolution for PINNs (Wong et al., 2025). Unlike linear scalarization, dominance-based evolutionary schemes can, in principle, cover concave and even disconnected Pareto regions, however their common drawback is heavy computational demands.

**Constrained formulations and hard constraints.** An alternative line replaces ad-hoc scalarization by explicit constraints. The following approaches have been applied to inverse problems in PINNs. hPINNs impose hard boundary/auxiliary constraints and use penalty/augmented-Lagrangian machinery within PINN training, targeting inverse design tasks (Lu et al., 2021). PECANN formulates forward/inverse learning as equality-constrained programs and introduces adaptive per-constraint ALM updates and minibatching (Basir & Senocak, 2022). ADMM-PINNs split PDE constraints from nonsmooth regularization, enabling proximal handling of sparsity/control objectives (Song et al., 2024). While these methods demonstrate strong empirical performance, their shared reliance on a nested optimization loop introduces significant computational overhead. This structure is fundamentally incompatible with the standard, single-loop paradigm of gradient-based optimizers used in deep learning.

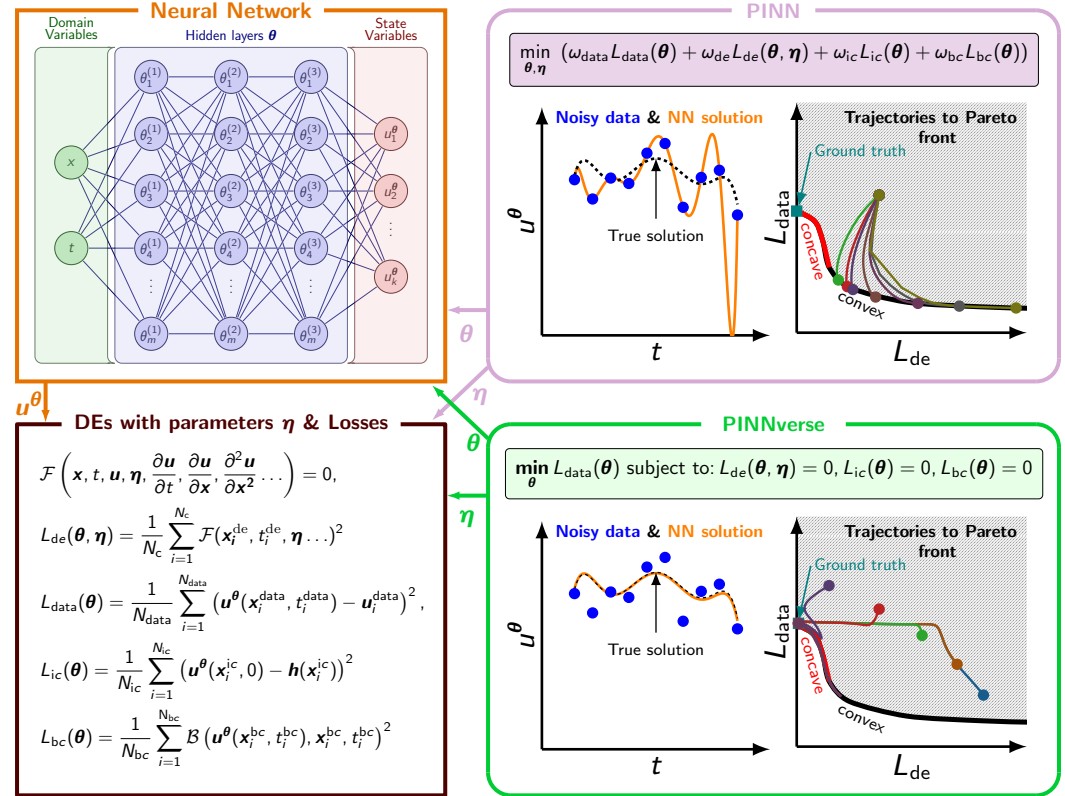

Figure 1: **Schematic comparison of PINN and PINNverse.** A neural network (top left) approximates the solution to a set of differential equations with unknown parameters $\boldsymbol{\eta}$ by minimizing several loss terms (bottom left). **Top Right (Standard PINN):** Minimizes a weighted sum of all losses. This approach struggles to find optimal trade-offs, as it is restricted to the convex regions of the Pareto front and requires tedious weight tuning. **Bottom Right (PINNverse):** Reformulates the problem as a constrained optimization, minimizing data loss subject to the physical constraints. This allows the optimizer to robustly converge to the ground truth solution by navigating the entire Pareto front and strictly enforcing physical laws.

## 3 METHODS

### 3.1 BACKGROUND: THE PINN MULTI-OBJECTIVE CHALLENGE

A Physics-Informed Neural Network (PINN) approximates the solution $\boldsymbol{u^\theta}(\boldsymbol{x}, t)$ of a differential equation system, where $\boldsymbol{\theta}$ are the network parameters. For inverse problems with unknown physical parameters $\boldsymbol{\eta}$, the network is trained by minimizing several competing loss functions (Figure 1):

- **Data Loss** ($L_{\text{data}}$): The discrepancy between the network's predictions and $N_{\text{data}}$ noisy observations, typically a root mean squared error.
- **Physics Loss** ($L_{\text{de}}, L_{\text{ic}}, L_{\text{bc}}$): The residuals of the governing differential equations, initial conditions, and boundary conditions, evaluated at a set of collocation points.

The standard approach combines these into a single objective via a weighted sum, $L_{\text{pinn}} = \sum_i \omega_i L_i$, which is then minimized with respect to both $\boldsymbol{\theta}$ and $\boldsymbol{\eta}$ using gradient descent (Raissi et al., 2019), as detailed in Appendix A.1. This formulation, however, is fundamentally limited. Training a PINN with noisy data is an inherent multi-objective optimization problem. The weighted-sum method can only converge to solutions on the convex hull of the Pareto front (Das & Dennis, 1997), a key concept we elaborate on in Appendix A.2. Furthermore, there is no principled way to set the weights $\omega_i$ a priori to target a specific solution on that convex front; achieving a desired balance between data-fit and physics-compliance requires extensive and often impractical hyperparameter tuning. Optimal solutions in non-convex regions remain systematically inaccessible (Figure 1).

## 3.2 PINNVERSE: A CONSTRAINED OPTIMIZATION FRAMEWORK

To overcome these limitations, we introduce PINNverse, which recasts the inverse problem as a constrained optimization task, a formulation we detail further in Appendix A.3. We minimize only the data loss, but subject to physics-based constraints:

$$
\begin{aligned}
\underset{\boldsymbol{\theta}, \boldsymbol{\eta}}{\text{minimize}} \quad & L_{\text{data}}(\boldsymbol{\theta}) \\
\text{subject to} \quad & L_i(\boldsymbol{\theta}, \boldsymbol{\eta}) = 0, \quad i \in \{\text{de}, \text{ic}, \text{bc}\} \\
\text{and} \quad & \boldsymbol{\eta}^{\text{lower}} \leq \boldsymbol{\eta} \leq \boldsymbol{\eta}^{\text{upper}}.
\end{aligned}
$$

This formulation is powerful because it removes the ambiguity of weight tuning and naturally integrates known parameter bounds in the optimization. It precisely defines the target on the Pareto front: the point that best fits the data while perfectly satisfying the physical laws. This allows convergence to any point on the front, regardless of its geometry.

We solve this problem using the Modified Differential Method of Multipliers (MDMM) (Platt & Barr, 1987), which optimizes the following augmented Lagrangian loss function

$$
\mathcal{L}_{\text{A}}(\boldsymbol{\theta}, \boldsymbol{\eta}, \boldsymbol{\lambda}) = L_{\text{data}}(\boldsymbol{\theta}) + \sum_i \left( \lambda_i L_i(\boldsymbol{\theta}, \boldsymbol{\eta}) + \frac{c_i}{2} L_i^2(\boldsymbol{\theta}, \boldsymbol{\eta}) \right),
$$

where $\lambda_i$ are Lagrange multipliers and $c_i > 0$ are penalty coefficients.

The key advantage of MDMM is its efficient, simultaneous update of all variables in a single step. Traditional augmented Lagrangian methods require a computationally expensive nested loop—fully optimizing primal variables $(\boldsymbol{\theta}, \boldsymbol{\eta})$ before taking a single update step on the dual variables $(\boldsymbol{\lambda})$—which is impractical for training neural networks. MDMM elegantly avoids this by updating the neural network parameters, differential equation parameters, and Lagrange multipliers all at once within a single backward pass. Appendix A.4 provides a full description of the update rules:

- **Primal update (Gradient Descent):** Update $\boldsymbol{\theta}$ and $\boldsymbol{\eta}$ to minimize $\mathcal{L}_{\text{A}}$.
- **Dual update (Gradient Ascent):** Update Lagrange multipliers $\boldsymbol{\lambda}$ to maximize $\mathcal{L}_{\text{A}}$.

This formulation allows the use of any gradient based optimizer normally used in neural network training. The quadratic penalty terms in $\mathcal{L}_{\text{A}}$ provide stability, acting as dampers that make this simultaneous update scheme converge robustly (Platt & Barr, 1987). This transforms the saddle-point problem into a tractable optimization with only a minor computational overhead compared to an unconstrained standard PINN.

## 4 EXPERIMENTS

### 4.1 EXPERIMENTAL DESIGN

To evaluate PINNverse, we benchmark it on a suite of four problems (two ODEs and two PDEs) against a standard PINN and the widely-used robust Nelder–Mead optimization algorithm (Nelder & Mead, 1965) from SciPy (Virtanen et al., 2020) to compare against a non PINN method. For a controlled comparison, PINNverse and the standard PINN share identical neural network architectures, initializations, and learning rate schedules, using the Adam optimizer (Kingma & Ba, 2014) for training. Complete training specifications are listed in Appendix A.5. This setup ensures that performance differences are attributable to our proposed training strategy. To isolate the performance gains of our method, we benchmark against a standard PINN using a simple weighted-sum loss where all weights are set to unity (Rohrhofer et al., 2021; Raissi et al., 2019). This ensures that the sole differentiating factor is the MDMM-based constrained optimization. We do not explore other weighting schemes for the baseline, as any static weighting would still be fundamentally restricted to the convex parts of the Pareto front without a principled way to select weights a priori.

We evaluate the performance across all three methods based on the relative error in the parameters $\beta$, the absolute and relative error on measured data for the estimated parameters plugged in a numerical solver $\mu_{\text{rel}}/\mu_{\text{abs}}$ and for PINN and PINNverse the maximum discrepancy of the learned neural network solution and the corresponding numerical solution $\gamma_{\text{ODE}}/\gamma_{\text{PDE}}$, see appendix A.6 for

more details. Across all benchmarks, PINNverse consistently demonstrates superior performance in parameter estimation and robustness to data noise and varied initial parameter guesses, as summarized in Figure 6. Moreover, PINNverse's computational overhead is minimal: while memory consumption is slightly elevated compared to a standard PINN, the runtime is highly competitive, falling within a 0.9–1.2× range (Figure 7). We attribute the occasional speedup (runtime ¡ 1.0×) to the more stable gradient updates inherent to our constrained optimization approach. Notably we observe that PINN only delivers good results in the case of no noise across all four experiments. This is expected as the Pareto front in this case collapses to a single point.

## 4.2 KINETIC REACTION MODEL

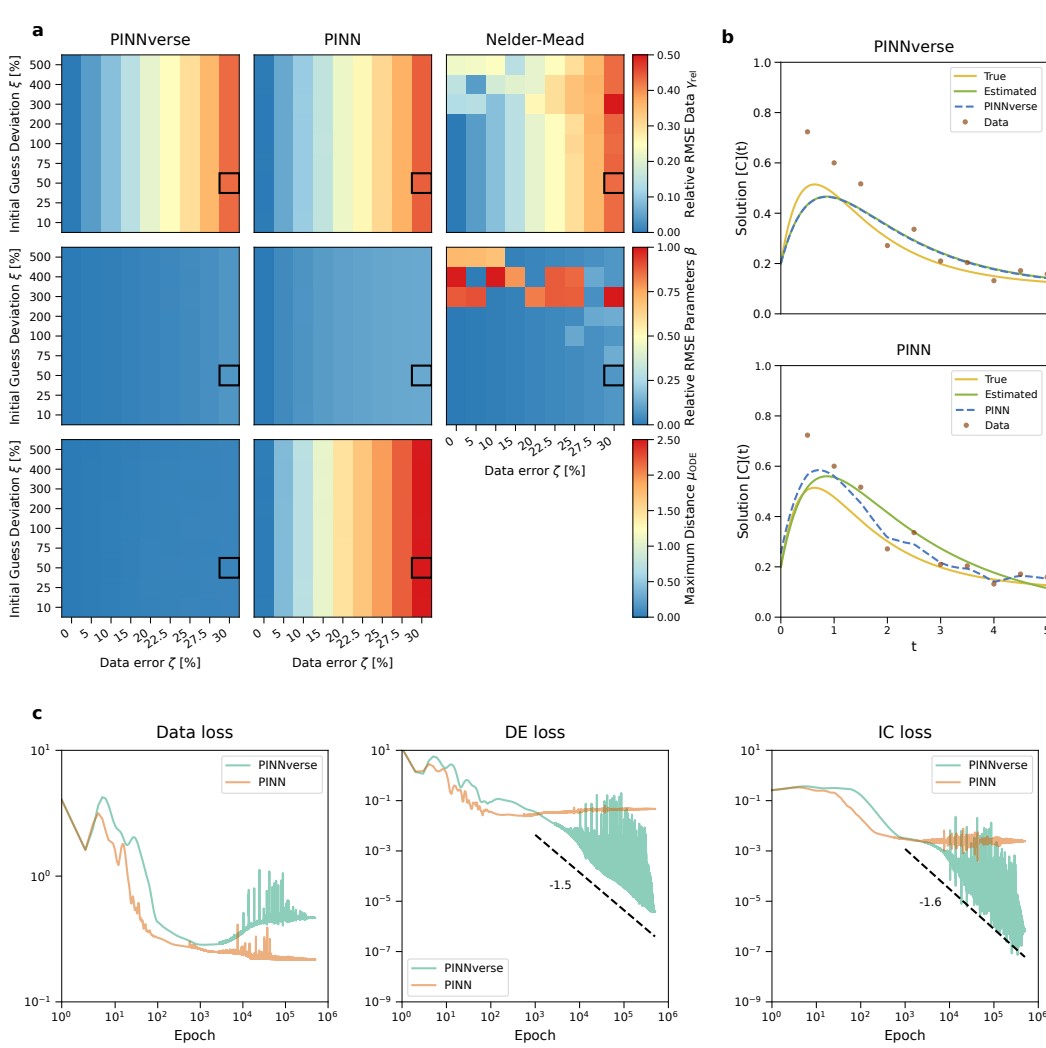

Figure 2: **Parameter estimation performance in the kinetic reaction ODE model. a**, Heatmaps depicting performance metrics across varying noise levels in the data, $\zeta$, and deviations in initial parameter guesses, $\xi$ (Methods). The black square highlights the scenario $\zeta = 30\%$, $\xi = 50\%$ analyzed in detail in subsequent panels. **b**, Comparison of trajectories for species $[C](t)$, generated using estimated parameters (green curve), true parameters (yellow curve), neural network predictions (blue curve) and the corresponding noisy observational data (brown dots). **c**, Training loss evolution for PINNverse and conventional PINN. Data, differential equation (DE) and initial condition (IC) losses are depicted. For PINNverse, a power law was fitted to the DE and IC losses after 1000 epochs (shifted dashed lines) with indicated exponents.

We first test our method on a nonlinear kinetic reaction model involving four species, previously used for parameter estimation with PINNs (Bibeau et al., 2024):

$$A \underset{k_2}{\overset{k_1}{\rightleftharpoons}} B + C, \quad C \underset{k_4}{\overset{k_3}{\rightleftharpoons}} D.$$

The system is described by four coupled ODEs with initial conditions $[A](0) = 1.0, [B](0) = 0, [C](0) = 0.2, [D](0) = 0$. We generate synthetic data at ten time points from true parameters $\boldsymbol{\eta}_{\text{true}} = [k_1, k_2, k_3, k_4] = [1.5, 0.5, 1, 0.1]$ and train for 500,000 epochs. As shown in Figure 2a, while all methods perform similarly on data-fit error, PINNverse excels in parameter estimation accuracy ($\beta$). It achieves a 4.1-fold mean improvement over the standard PINN and remains robust to poor initial guesses where Nelder–Mead fails (e.g., $\xi = 500\%$). The standard PINN overfits noisy data, leading to physically inaccurate solutions, whereas PINNverse remains physics-conforming (Figure 2b). This is reflected in the maximum ODE deviation ($\mu_{\text{ODE}}$), where PINNverse shows a 33-fold improvement over the standard PINN in noisy scenarios (Figure 2a, bottom row). The loss curves (Figure 2c) confirm that the standard PINN prioritizes the data term at the expense of the physics and initial condition loss after an initial phase. In contrast, PINNverse does not overfit on the data loss but rather shows superlinear algebraic convergence ($L \sim \text{epoch}^{-a}$, $a > 1$) with the physics and initial condition loss.

## 4.3 FITZHUGH–NAGUMO MODEL

Next, we use the FitzHugh–Nagumo (FHN) model (R., 1961; Nagumo et al., 1962), a classic system exhibiting excitable dynamics often used as a PINN benchmark (Rudi et al., 2021; Bizzi et al., 2025):

$$\frac{\mathrm{d}u}{\mathrm{d}t} = u - \frac{u^3}{3} - v, \quad \frac{\mathrm{d}v}{\mathrm{d}t} = \frac{u + a - bv}{r},$$

with initial conditions $u(0) = 0, v(0) = 0$. For a challenging sparse-data scenario, we use only ten data points generated from $\boldsymbol{\eta}_{\text{true}} = [a, b, r] = [0.7, 0.8, 12.5]$ and train for 500,000 epochs. Figure 3a shows that PINNverse achieves a 11.6-fold and 13.5-fold mean improvement in data fit ($\gamma_{\text{rel}}$) and parameter estimation ($\beta$), respectively, compared to the standard PINN. Nelder–Mead again fails for poor initial guesses ($\xi = 500\%$), where PINNverse remains robust. As before, the standard PINN overfits noisy data, while PINNverse's solution remains physically consistent (Figure 3b). The loss curves (Figure 3c) show the standard PINN's failure to reduce the physics loss after $\approx 1000$ epochs, while PINNverse convergences algebraically for all physical loss terms.

## 4.4 FISHER–KPP MODEL

Our first PDE benchmark is the Fisher–KPP equation (Fisher, 1937; Kolmogorov et al., 1937), which models reaction-diffusion phenomena and admits challenging traveling wave solutions:

$$\frac{\partial u}{\partial t} = D \frac{\partial^2 u}{\partial x^2} + \rho u(1 - u),$$

$$u(x, 0) = \frac{1}{10} e^{-x}, \quad \frac{\partial u(x, t)}{\partial x} = 0 \quad \text{at} \quad x \in \{0, 10\}.$$

We use 18 data points generated from $\boldsymbol{\eta}^{\text{true}} = [D, \rho] = [0.5, 1]$ and train for 300,000 epochs. Figure 4a shows that PINNverse substantially outperforms the standard PINN in noisy settings, achieving 5-fold and 10-fold mean improvements in data fit ($\gamma_{\text{abs}}$) and parameter accuracy ($\beta$), respectively. The standard PINN's poor parameter estimates yield numerical solutions that fail to approximate the data, even as the network itself overfits the noisy measurements (Figure 4b). PINNverse algebraically minimizes all physical loss components, unlike the standard PINN (Figure 4c).

## 4.5 BURGERS' EQUATION

Our final benchmark is the viscous Burgers' equation (Burgers, 1948), a standard test for PINNs (Raissi et al., 2019) that models nonlinear advection and diffusion capable of forming sharp shocks:

$$\frac{\partial u}{\partial t} + u \frac{\partial u}{\partial x} = \nu \frac{\partial^2 u}{\partial x^2},$$

$$u(0, x) = -\sin(\pi x), \quad u(t, x) = 0 \quad \text{at} \quad x \in \{-1, 1\}.$$

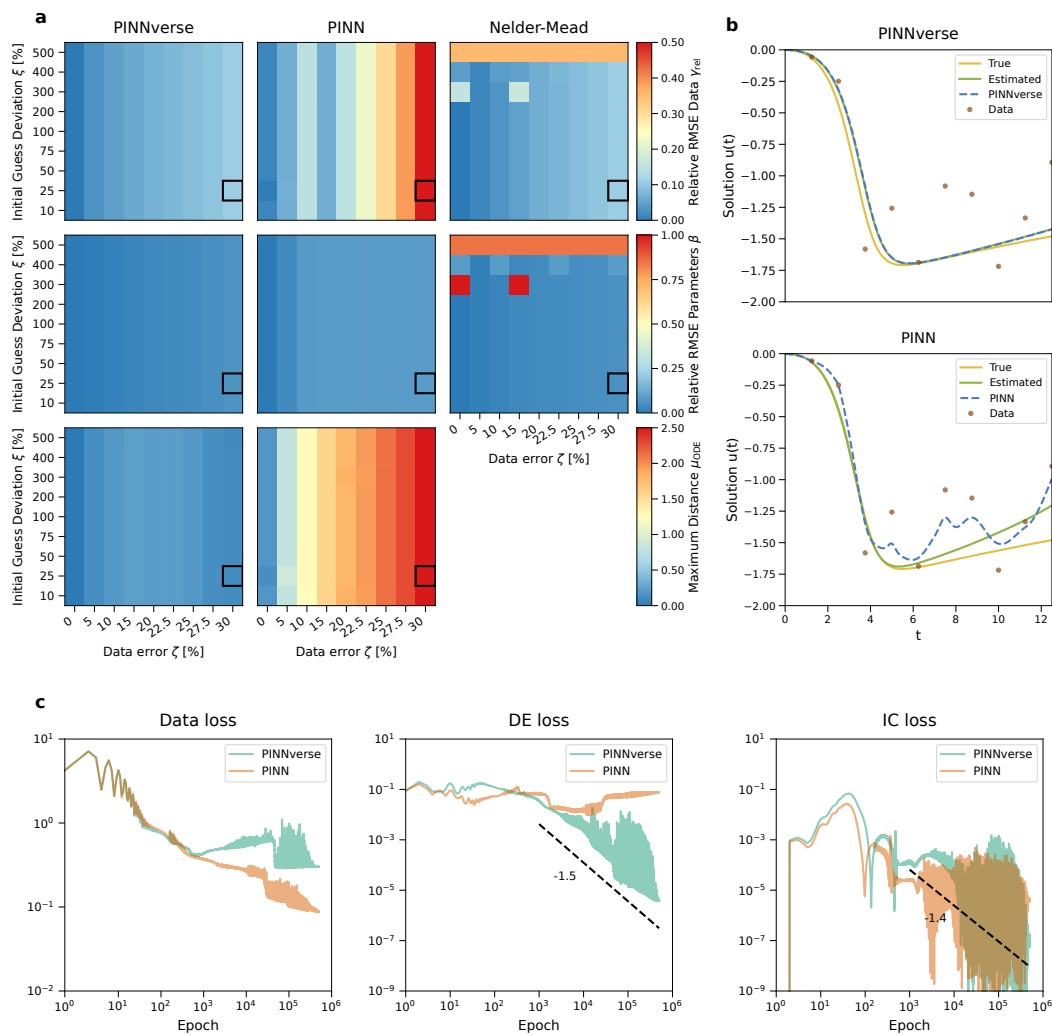

Figure 3: **Parameter estimation performance in the FitzHugh–Nagumo ODE model.** **a**, Heatmaps depicting performance metrics across varying noise levels in the data, $\zeta$, and deviations in initial parameter guesses, $\xi$ (Methods). The black square highlights the scenario $\zeta = 30\%, \xi = 25\%$ analyzed in detail in subsequent panels. **b**, Comparison of trajectories for the excitable variable $u(t)$, generated using estimated parameters (green curve), true parameters (yellow curve), neural network predictions (blue curve), and the corresponding noisy observational data (brown dots). **c**, Training loss evolution for PINNverse and conventional PINN. Data, differential equation (DE) and initial condition (IC) losses are depicted. For PINNverse, a power law was fitted to the DE and IC losses after 1000 epochs (shifted dashed lines) with indicated exponents.

We set the viscosity $\eta^{\text{true}} = \nu = 0.01$ to operate in the challenging shock wave regime, use 12 data points, and train for 150,000 epochs. To handle the high-frequency solution, we apply a Fourier feature mapping to the spatial input (Tancik et al., 2020). While all methods achieve similar data-fit RMSE (Figure 5a, top row), PINNverse's parameter estimation is substantially more accurate. It achieves a 54-fold improvement in $\beta$ over the standard PINN, which only performs well in the noise-free case, and a 19-fold improvement over Nelder–Mead. While the standard PINN over-fits, PINNverse accurately captures the shock structure even with noisy data (Figure 5b) and again consistently minimizes all physical loss terms with algebraic convergence (Figure 5c).

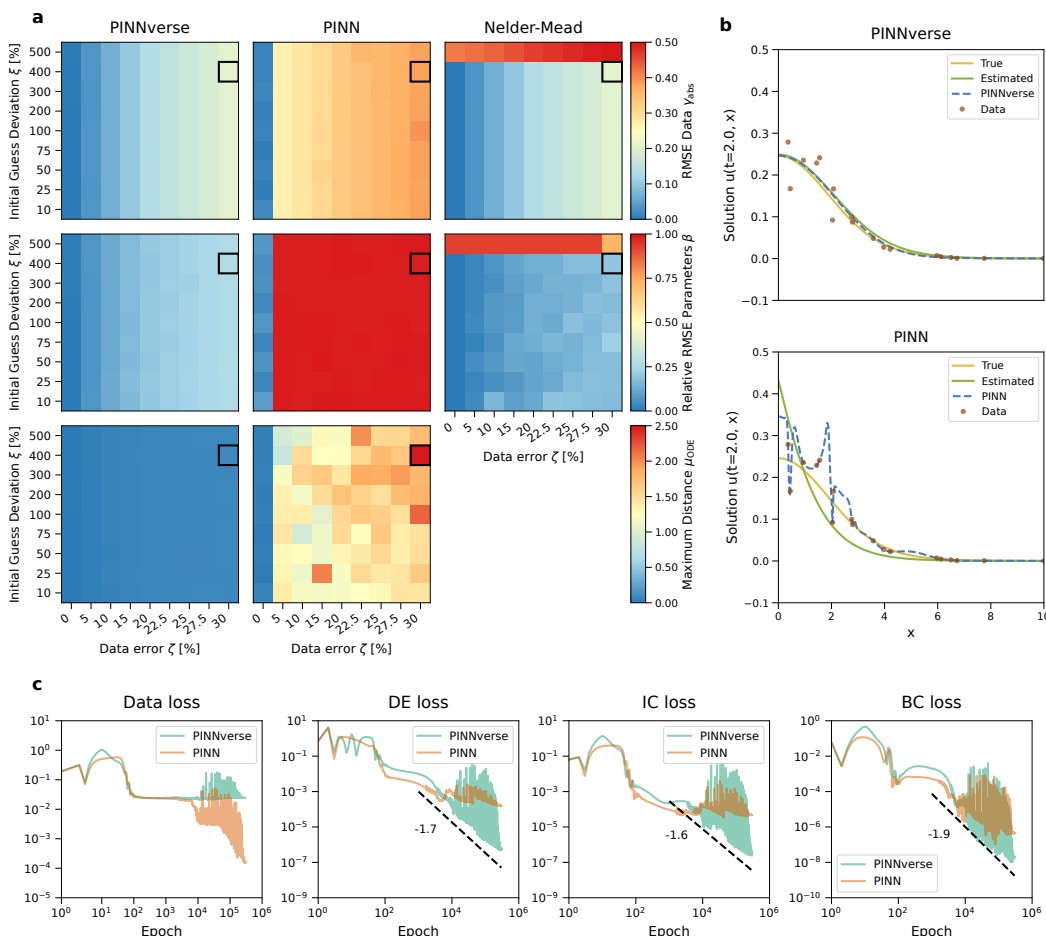

Figure 4: **Parameter estimation performance in the Fisher–KPP PDE model. a**, Heatmaps depicting performance metrics across varying noise levels in the data, $\zeta$, and deviations in initial parameter guesses, $\xi$ (Methods). The black square highlights the scenario $\zeta = 30\%$, $\xi = 400\%$ analyzed in detail in subsequent panels. **b**, Comparison of trajectories for the cell concentration $u(x)$ at time point $t = 2$, generated using estimated parameters (green curve), true parameters (yellow curve), neural network predictions (blue curve), and the corresponding noisy observational data (brown dots). **c**, Training loss evolution for PINNverse and conventional PINN. Data, differential equation (DE), initial condition (IC) and boundary condition (BC) losses are depicted. For PINNverse, a power law was fitted to the DE, IC and BC losses after 1000 epochs (shifted dashed lines) with indicated exponents.

## 5 DISCUSSION

Standard Physics-Informed Neural Networks often falter in parameter inference from noisy data. Their reliance on a static, weighted-sum loss function is a brittle approach for navigating the complex trade-off between data fidelity and physical consistency (Cai et al., 2022; Cuomo et al., 2022). This can lead to convergence at undesirable points on the Pareto front, where the learned solution overfits the data or deviates substantially from the governing equations.

We introduce PINNverse, which recasts the PINN inverse problem as a constrained optimization task solved with the Modified Differential Method of Multipliers (MDMM). This enables convergence to any point on the Pareto front, overcoming the convex-hull limitation of traditional weighted-sum methods. Experimentally, while baseline PINNs overfit noisy data and yield nonphysical solutions, PINNverse accurately recovers the true underlying dynamics. This was particularly stark for Burgers' equation, where PINNverse captured shock waves and achieved superlinear convergence of the physics loss, a trait absent in the baseline.

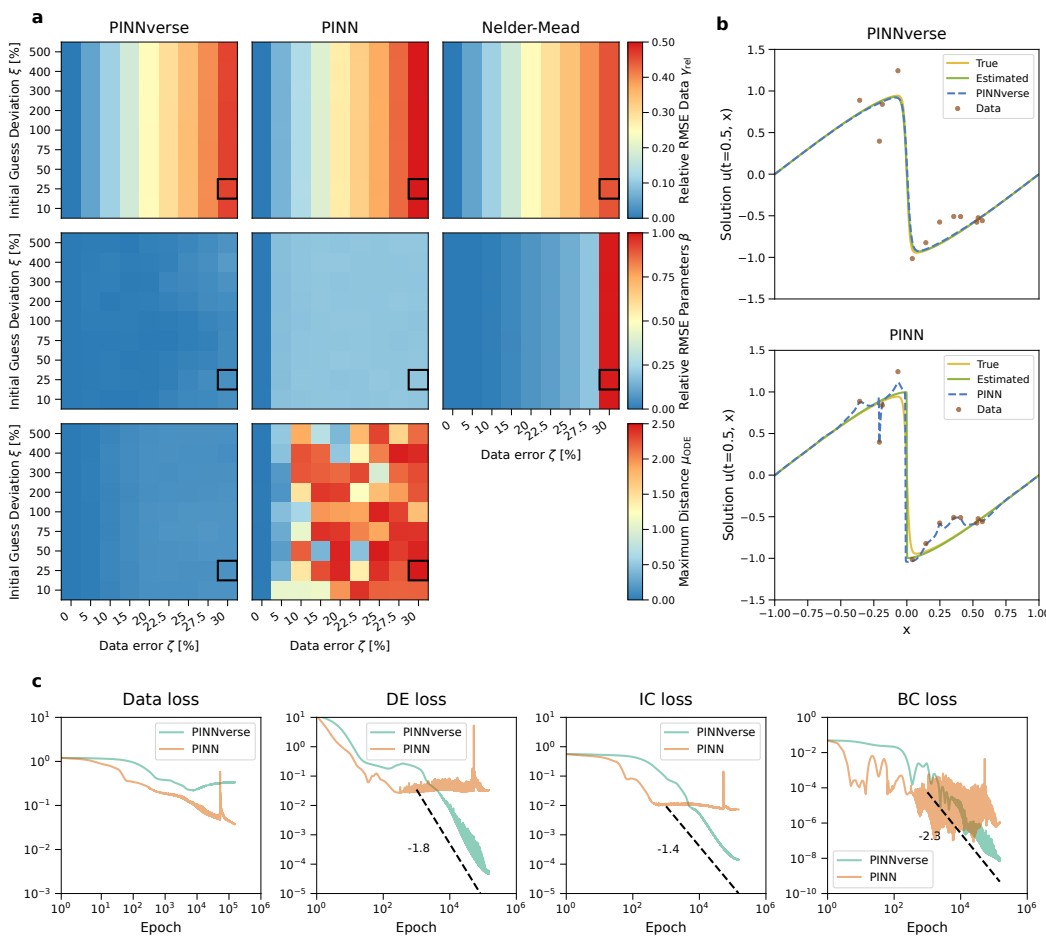

Figure 5: **Parameter estimation performance in Burgers' PDE model. a**, Heatmaps depicting performance metrics across varying noise levels in the data, $\zeta$, and deviations in initial parameter guesses, $\xi$ (Methods). The black square highlights the scenario $\zeta = 30\%$, $\xi = 25\%$ analyzed in detail in subsequent panels. **b**, Comparison of trajectories for the dependent variable $u(x)$ at time point $t = 0.5$, generated using estimated parameters (green curve), true parameters (yellow curve), neural network predictions (blue curve), and the corresponding noisy observational data (brown dots). **c**, Training loss evolution for PINNverse and conventional PINN. Data, differential equation (DE), initial condition (IC) and boundary condition (BC) losses are depicted. For PINNverse, a power law was fitted after 1000 epochs to the DE, IC and BC losses (shifted dashed lines) with indicated exponents.

Beyond its theoretical strengths, PINNverse offers significant practical benefits. It is more robust to initial parameter guesses than classical optimizers like Nelder–Mead and elegantly incorporates parameter bounds as simple constraints. Remarkably, these advantages are realized with minimal code modification to an existing PINN framework and only a minor increase in computational cost.

While PINNverse relies on hyperparameter tuning like other deep learning methods, our study found it to be resilient even without extensive optimization. The modularity of our approach is a key strength, inviting future work that integrates PINNverse with other state-of-the-art PINN enhancements for forward problems, such as causal training (Wang et al., 2024) or adaptive resampling (Wu et al., 2022; Daw et al., 2023). Replacing the MLP with Kolmogorov Arnold neural networks (Liu et al., 2025) could also yield performance increase. Furthermore, a rigorous quantitative analysis comparing its computational complexity against traditional numerical solvers in high-dimensional settings presents an exciting research avenue. In conclusion, PINNverse provides a robust and principled framework for solving physics-informed inverse problems, eliminating a core limitation of the standard PINN methodology.

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

# A APPENDIX

## A.1 SOLVING THE INVERSE PROBLEM WITH PHYSICS-INFORMED NEURAL NETWORKS

We consider a general differential equation (DE) system represented in a residual form given by

$$\mathcal{F}(\boldsymbol{x}, t, \boldsymbol{u}, \boldsymbol{\eta}, \boldsymbol{u}_t, \nabla \boldsymbol{u}, ...) = 0, \quad \boldsymbol{x} \in \Omega, \quad t \in [0, T]$$
$$\mathcal{B}(\boldsymbol{u}(\boldsymbol{x}, t), \boldsymbol{x}, t) = 0, \quad \boldsymbol{x} \in \partial\Omega, \ t \in [0, T]$$
$$\boldsymbol{u}(\boldsymbol{x}, 0) = \boldsymbol{h}(\boldsymbol{x}), \quad \boldsymbol{x} \in \Omega$$

where $\Omega \subseteq \mathbb{R}^n$ represents the spatial domain with boundary $\partial\Omega$, and $\boldsymbol{u} : \Omega \times [0, T] \to \mathbb{R}^m$ denotes the solution field over the space-time domain. The operator $\mathcal{F}(\cdot)$ is a spatio-temporal differential operator encapsulating the governing physics of the system, which may incorporate multiple parameters $\boldsymbol{\eta} \in \mathbb{R}^p$ and various spatial and temporal derivatives of $\boldsymbol{u}$. The boundary conditions are imposed through the spatio-temporal operator $\mathcal{B}(\boldsymbol{u}, \boldsymbol{x}, t)$, which acts on the solution at the domain boundary $\partial\Omega$. The initial solution $\boldsymbol{h}(\boldsymbol{x})$ prescribes the state of the system at time $t = 0$ throughout $\Omega$.

The forward problem consists of determining the solution $\boldsymbol{u}(\boldsymbol{x}, t; \boldsymbol{\eta})$, given known parameters $\boldsymbol{\eta}$. The inverse problem regards the parameters $\boldsymbol{\eta}$ as unknown quantities, requiring inference from observational data at discrete spatio-temporal locations. To formalize this inverse scenario, we assume the availability of a dataset comprising $N_{\text{data}}$ observations:

$$\{(\boldsymbol{x}_i^{\text{data}}, t_i^{\text{data}}, \boldsymbol{u}_i^{\text{data}})\}_{i=1}^{N_{\text{data}}},$$

where each datum consists of a coordinate pair $(\boldsymbol{x}_i^{\text{data}}, t_i^{\text{data}})$ and the corresponding observed solution $\boldsymbol{u}_i^{\text{data}}$. The goal is to approximate the solution using a NN model $\boldsymbol{u}^{\boldsymbol{\theta}}(\boldsymbol{x}, t)$, parameterized by neuronal weights and biases collectively denoted by $\boldsymbol{\theta}$. The optimal parameters are determined by minimizing the discrepancy between the network predictions and observed data in some chosen metric:

$$\boldsymbol{\theta}^* = \arg\min_{\boldsymbol{\theta}} L_{\text{data}}(\boldsymbol{\theta}).$$

We seek the minimum in a least-squares sense here. The data loss function $L_{\text{data}}(\boldsymbol{\theta})$ can be formulated as either an absolute loss function

$$L_{\text{data}}(\boldsymbol{\theta}) = \sqrt{\frac{1}{N_{\text{data}}} \sum_{i=1}^{N_{\text{data}}} \left(\boldsymbol{u}^{\boldsymbol{\theta}}(\boldsymbol{x_i}, t_i) - \boldsymbol{u}_i^{\text{data}}\right)^2}$$

or, unless the data approaches zero, as a relative loss function

$$L_{\text{data}}(\boldsymbol{\theta}) = \sqrt{\frac{1}{N_{\text{data}}} \sum_{i=1}^{N_{\text{data}}} \left(\frac{\boldsymbol{u}^{\boldsymbol{\theta}}(\boldsymbol{x_i}, t_i) - \boldsymbol{u}_i^{\text{data}}}{\boldsymbol{u}_i^{\text{data}}}\right)^2}$$

where the vector division is taken element-wise.

To impose conformity with the physical laws described by the DE, a residual loss is introduced as

$$L_{\text{de}}(\boldsymbol{\theta}, \boldsymbol{\eta}) = \frac{1}{N_c} \sum_{i=1}^{N_c} \mathcal{F}(\boldsymbol{x}_i^{\text{de}}, t_i^{\text{de}}, \boldsymbol{u}^{\theta}(\boldsymbol{x}_i^{\text{de}}, t_i^{\text{de}}), \boldsymbol{\eta}, \ldots)^2,$$

where $\{\boldsymbol{x}_i^{\text{de}}, t_i^{\text{de}}\}_{i=1}^{N_c}$ are collocation points sampled within $\Omega \times (0, T)$.

Additionally, losses for initial and boundary conditions are defined by

$$L_{\text{bc}}(\boldsymbol{\theta}) = \frac{1}{N_{\text{bc}}} \sum_{i=1}^{N_{\text{bc}}} \mathcal{B}\left(\boldsymbol{u}^{\boldsymbol{\theta}}(\boldsymbol{x}_i^{\text{bc}}, t_i^{\text{bc}}), \boldsymbol{x}_i^{\text{bc}}, t_i^{\text{bc}}\right)^2$$

$$L_{\text{ic}}(\boldsymbol{\theta}) = \frac{1}{N_{\text{ic}}} \sum_{i=1}^{N_{\text{ic}}} \left(\boldsymbol{u}^{\boldsymbol{\theta}}(\boldsymbol{x}_i^{\text{ic}}, 0) - \boldsymbol{h}(\boldsymbol{x}_i^{\text{ic}})\right)^2$$

where $\{\boldsymbol{x}_i^{\mathrm{bc}}, t_i^{\mathrm{bc}}\}_{i=1}^{N_{\mathrm{bc}}}$ are boundary condition points in $\partial\Omega \times [0, T]$ and $\{\boldsymbol{x}_i^{\mathrm{ic}}, 0\}_{i=1}^{N_{\mathrm{ic}}}$ are initial conditional points in $\Omega \times \{t = 0\}$.

Traditionally, a composite total loss function is then formulated as Raissi et al. (2019)

$$L_{\mathrm{pinn}}(\boldsymbol{\theta}, \boldsymbol{\eta}) = \omega_{\mathrm{data}} L_{\mathrm{data}}(\boldsymbol{\theta}) + \omega_{\mathrm{de}} L_{\mathrm{de}}(\boldsymbol{\theta}, \boldsymbol{\eta})$$
$$+ \omega_{\mathrm{ic}} L_{\mathrm{ic}}(\boldsymbol{\theta}) + \omega_{\mathrm{bc}} L_{\mathrm{bc}}(\boldsymbol{\theta})$$

where $\omega_{\mathrm{data}}$, $\omega_{\mathrm{de}}$, $\omega_{\mathrm{ic}}$ and $\omega_{\mathrm{bc}}$ are weights that balance the partial losses. Following common practice Raissi et al. (2019), all weights were set to one here.

PINNs leverage automatic differentiation to compute derivatives of the output variables $\boldsymbol{u}$ with respect to $\boldsymbol{x}$ and $t$, enabling evaluation of the differential operators $\mathcal{F}(\cdot)$ and boundary operators $\mathcal{B}(\cdot)$. The parameter update using gradient descent is performed as

$$\boldsymbol{\theta}^{(k+1)} = \boldsymbol{\theta}^{(k)} - \alpha \nabla_{\boldsymbol{\theta}} L_{\mathrm{pinn}}(\boldsymbol{\theta}^{(k)}, \boldsymbol{\eta}^{(k)})$$
$$\boldsymbol{\eta}^{(k+1)} = \boldsymbol{\eta}^{(k)} - \alpha \nabla_{\boldsymbol{\eta}} L_{\mathrm{pinn}}(\boldsymbol{\theta}^{(k)}, \boldsymbol{\eta}^{(k)})$$

where $\alpha > 0$ is the learning rate and $k$ the iteration index.

## A.2 PARETO OPTIMALITY

With noisy experimental data, the composite loss function encompasses multiple competing objectives. This represents a multi-objective optimization problem

$$\min_{\Psi} \boldsymbol{L}(\boldsymbol{\Psi}).$$

where we seek to simultaneously optimize all components of a total loss vector

$$\boldsymbol{L}(\boldsymbol{\Psi}) = \boldsymbol{L}(\boldsymbol{\theta}, \boldsymbol{\eta})$$
$$= (L_{\mathrm{data}}(\boldsymbol{\theta}), L_{\mathrm{de}}(\boldsymbol{\theta}, \boldsymbol{\eta}), L_{\mathrm{ic}}(\boldsymbol{\theta}), L_{\mathrm{bc}}(\boldsymbol{\theta}))^{\mathrm{T}}.$$

Finding a single parameter vector $\boldsymbol{\Psi}$ that simultaneously minimizes all loss components is generally infeasible. To formalize this challenge, we adopt the concept of Pareto optimality. A parameter vector $\boldsymbol{\Psi}$ is considered (globally) Pareto optimal, if no other parameter vector $\boldsymbol{\Psi}$ exists that achieves non-increasing values across all loss functions while strictly improving at least one loss component. The collection of all candidate solutions constitutes the feasible region.

The subset of optimal objective function values represents the Pareto front Pareto (1971 (1906) (Fig. 1, solid black curve), which manifests in two fundamental geometric configurations: convex and concave. A convex Pareto front is distinguished by the property that for any two points $\boldsymbol{a}$ and $\boldsymbol{b}$ on the front and any scalar $\kappa \in [0, 1]$, there exists a point $\boldsymbol{c}$ on the front such that $\kappa||\boldsymbol{a}|| + (1 - \kappa)||\boldsymbol{b}|| \geq ||\boldsymbol{c}||$. Conversely, a concave Pareto front satisfies the inequality $\kappa||\boldsymbol{a}|| + (1-\kappa)||\boldsymbol{b}|| \leq ||\boldsymbol{c}||$.

The performance of gradient-based optimization methods is intrinsically linked to the geometry of the Pareto front. Specifically, when minimizing linearly weighted objectives, gradient descent converges exclusively to solutions located on the convex regions of the Pareto front Das & Dennis (1997). Consequently, regardless of the positive weighting parameters selected, points within non-convex segments of the front cannot be attained, as they do not correspond to minima of any weighted sum objective function. In contrast, for purely convex Pareto fronts, gradient-based optimization can theoretically converge to any desired point along the curve through appropriate adjustment of the weighting parameters. However, precisely controlling the final solution point along the front via weight selection is often non-trivial, as the mapping between weights and Pareto points is highly sensitive to the front's local curvature Das & Dennis (1997).

In practical applications with neural networks, Pareto fronts typically have mixed shapes with both convex and concave regions, making the tuning of PINNs notoriously difficult Wong et al. (2025).

## A.3 INVERSE PROBLEM AS CONSTRAINT OPTIMIZATION

To address the limitations of standard gradient-based methods on complex Pareto fronts, we reformulate the PINN training process as a constrained optimization problem. Rather than treating all

loss terms equally, we designate the data-fitting term as the primary objective while transforming the physics-based terms into constraints:

$$\underset{\boldsymbol{\theta}}{\text{minimize}} \quad L_{\text{data}}(\boldsymbol{\theta})$$

$$\text{subject to} \quad L_i(\boldsymbol{\theta}, \boldsymbol{\eta}) = 0, \quad i \in \mathcal{I}_{\text{e}} = \{\text{de}, \text{ic}, \text{bc}\}$$

$$\eta_j^{\text{lower}} \leq \eta_j \leq \eta_j^{\text{upper}}, \quad j \in \mathcal{I}_{\text{b}} = \{1, \dots, p\}$$

The parameters $\eta_j \in$ represent differential equation parameters constrained within physically plausible bounds $[\eta_j^{\text{lower}}, \eta_j^{\text{upper}}]$. These bounds ensure that the solution remains physically meaningful and prevent the neural network from exploring invalid regions.

To handle the bound constraints efficiently, we introduce an infeasibility function

$$V_j(\eta_j(\boldsymbol{\theta})) = \max(\eta_j^{\text{lower}}, \min(\eta_j(\boldsymbol{\theta}), \eta_j^{\text{upper}})) - \eta_j(\boldsymbol{\theta})$$

that measures constraint violations. This allows us to express the Lagrangian as

$$\mathcal{L}(\boldsymbol{\theta}, \boldsymbol{\eta}, \boldsymbol{\lambda}, \boldsymbol{\chi}) = L_{\text{data}}(\boldsymbol{\theta}) + \sum_{i \in \mathcal{I}_e} \lambda_i L_i(\boldsymbol{\theta}, \boldsymbol{\eta})$$

$$+ \sum_{j \in \mathcal{I}_b} \chi_j V_j(\eta_j),$$

where $\lambda_i$ and $\chi_j$ represent the Lagrange multipliers of equality and parameter bound constraints. The optimal set of neural network parameters is then obtained through a min-max formulation:

$$(\boldsymbol{\theta}, \boldsymbol{\eta})^* = \arg \min_{\boldsymbol{\theta}, \boldsymbol{\eta}} \left( \max_{\boldsymbol{\lambda} \geq 0, \boldsymbol{\chi} \geq 0} \mathcal{L}(\boldsymbol{\theta}, \boldsymbol{\eta}, \boldsymbol{\lambda}, \boldsymbol{\chi}) \right).$$

The target solution for the Lagrangian min-max formulation is inherently a saddle point Boyd & Vandenberghe (2004). However, such points are generally not attractors for standard gradient-based optimizers Platt & Barr (1987).

### A.4 OPTIMIZATION APPROACH OF PINNVERSE

To ultimately overcome these limitations, we employ the Modified Differential Method of Multipliers (MDMM) Platt & Barr (1987). To the best of our knowledge, this represents the first application of the MDMM in the context of PINNs. MDMM is an optimization algorithm derived from the augmented Lagrangian formulation, also known as the Method of Multipliers. This formulation introduces quadratic penalty terms alongside the standard Lagrange multiplier terms to improve convergence properties. For the PINNverse problem, the augmented Lagrangian is defined as

$$\mathcal{L}_{\text{A}}(\boldsymbol{\theta}, \boldsymbol{\eta}, \boldsymbol{\lambda}, \boldsymbol{\chi}, \boldsymbol{c}) = L_{\text{data}}(\boldsymbol{\theta})$$

$$+ \sum_{i \in \mathcal{I}_e} \left( \lambda_i L_i(\boldsymbol{\theta}, \boldsymbol{\eta}) + \frac{c_i}{2} L_i^2(\boldsymbol{\theta}, \boldsymbol{\eta}) \right)$$

$$+ \sum_{j \in \mathcal{I}_b} \left( \chi_j V_j(\eta_j) + \frac{d_j}{2} V_j^2(\eta_j) \right),$$

where $c_i > 0$ and $d_j > 0$ are the penalty coefficients for the constraints. Larger values enforce constraints more strictly. In this study, all penalty parameters were set to unity ($c_i = d_j = 1$).

A key distinction of MDMM from standard sequential augmented Lagrangian methods lies in its update dynamics. MDMM proposes simultaneous updates for both the primal variables ($\boldsymbol{\theta}, \boldsymbol{\eta}$) and the Lagrange multipliers ($\boldsymbol{\lambda}, \boldsymbol{\chi}$). For a gradient descent update this reads

$$\boldsymbol{\theta}^{(k+1)} = \boldsymbol{\theta}^{(k)} - \alpha \nabla_{\boldsymbol{\theta}} \mathcal{L}_{\text{A}}(\boldsymbol{\theta}^{(k)}, \boldsymbol{\eta}^{(k)}, \boldsymbol{\lambda}^{(k)}, \boldsymbol{\chi}^{(k)})$$

$$\boldsymbol{\eta}^{(k+1)} = \boldsymbol{\eta}^{(k)} - \alpha \nabla_{\boldsymbol{\eta}} \mathcal{L}_{\text{A}}(\boldsymbol{\theta}^{(k)}, \boldsymbol{\eta}^{(k)}, \boldsymbol{\lambda}^{(k)}, \boldsymbol{\chi}^{(k)}).$$

Here $\alpha > 0$ represents the learning rate that controls the step size during each iteration of the gradient descent. Crucially, in MDMM the Lagrange multipliers are updated via gradient ascent:

$$\lambda_i^{(k+1)} = \lambda_i^{(k)} + \alpha \, L_i(\boldsymbol{\theta}^{(k)}, \boldsymbol{\eta}^{(k)}), \quad i \in \mathcal{I}_{\text{e}}$$

$$\chi_j^{(k+1)} = \chi_j^{(k)} + \alpha \, V_j(\eta_j^{(k)}), \quad j \in \mathcal{I}_{\text{b}}$$

The inclusion of the quadratic penalty term, governed by $c_i$, $d_j$, is essential. As established in optimization theory Nocedal & Wright (2006); Platt & Barr (1987), for sufficiently large penalty parameters, the Hessian of the augmented Lagrangian with respect to the primal variables ($\nabla_{\boldsymbol{\theta},\boldsymbol{\xi}}^2 \mathcal{L}_A$) becomes positive definite in the subspace tangent to the constraints near a constrained minimum satisfying standard second-order sufficiency conditions. This induces local convexity and transforms the constrained minimum into an attractor for the dynamics, mitigating the saddle-point issues associated with the standard Lagrangian that hinder simple gradient descent Platt & Barr (1987). Note that we still need gradient ascent for the Lagrange multipliers, since the convexity only holds for the primal variables.

Consequently, MDMM offers robust convergence towards a constrained minimum for sufficiently large $c_i$ and $d_j$, suitable learning rate $\alpha$, and initialization within the basin of attraction. Notably, this minimum can be any point on the Pareto front, even in the non-convex region.

In the theoretical derivation presented above, gradient updates were illustrated using stochastic gradient descent for simplicity. However, any gradient-based optimization algorithm is compatible with the MDMM framework. Motivated by this flexibility, we adopt the established Adam optimizer Kingma & Ba (2014).

### A.5 TRAINING REGIME

For all presented results, both the standard PINN and PINNverse were trained using neural networks comprising two hidden layers, each consisting of 20 neurons, with hyperbolic tangent activation functions. We employed a learning rate scheduler characterized by an initial linear decay from $\alpha = 10^{-2}$ down to $10^{-4}$ until reaching the last 30,000 epochs, after which the learning rate was kept constant at $\alpha = 10^{-4}$. An exception is the FitzHugh–Nagumo model, where we started with a learning rate of $\alpha = 5 \times 10^{-3}$. For discretization, $N_{\text{de}} = 16,384$ collocation points were uniformly distributed across the interior of the temporal or spatio-temporal domains using a Sobol sequence (Sobol, 1967). In the two PDEs, additional collocation points, specifically $N_{\text{ic}} = N_{\text{bc}} = 1,024$, were allocated to enforce the initial and boundary conditions, respectively.

For Burgers' equation, Fourier features (Tancik et al., 2020) were used in the training. This technique transforms the coordinate into a higher-dimensional feature space using sinusoidal basis functions. Ten such basis functions corresponding to distinct frequencies were employed in our network. This augmented spatial representation, concatenated with the temporal coordinate, served as the network input, thereby enhancing its capability to resolve the sharp gradients of shock wave dynamics.

### A.6 EVALUATION OF ACCURACY

We evaluated method performance under realistic conditions by introducing heteroscedastic Gaussian noise to the data,

$$\hat{y} \sim \mathcal{N}(y, \zeta y)$$

with noise levels $\zeta$ up to 30%. Additionally, we used substantially perturbed initial guesses for parameter initialization,

$$\boldsymbol{\eta}^{\text{start}} = (1 + \xi)\boldsymbol{\eta}^{\text{true}}$$

with relative deviations $\xi$ up to 500%.

To quantitatively evaluate solution accuracy, we define the maximum distance metric $\mu$. For ODE problems, this metric is formulated as

$$\mu_{\text{ODE}} = \max_{\substack{t \in [0,T] \\ i \in \{1,...,m\}}} \left| u_i^{\text{NN}}(t; \boldsymbol{\theta}, \boldsymbol{\eta}) - u_i^{\text{true}}(t; \boldsymbol{\eta}^{\text{true}}) \right|$$

where $u_i^{\text{NN}}(t; \boldsymbol{\theta})$ the neural network prediction with parameters $\boldsymbol{\theta}$ for the $i$-th solution component at time $t$, and $u_i^{\text{true}}(t; \boldsymbol{\eta}^{\text{true}})$ denotes the corresponding true solution with parameters $\boldsymbol{\eta}^{\text{true}}$ obtained via high-precision numerical methods. For PDE problems, we extend this metric to incorporate spatial dimensions:

$$\mu_{\text{PDE}} = \max_{\substack{t \in \mathcal{T} \\ \boldsymbol{x} \in \Omega \\ i \in \{1,...,m\}}} \left| u_i^{\text{NN}}(t, \boldsymbol{x}; \boldsymbol{\theta}, \boldsymbol{\eta}) - u_i^{\text{true}}(t, \boldsymbol{x}; \boldsymbol{\eta}^{\text{true}}) \right|$$

where $\Omega$ is the spatial domain and $\mathcal{T}$ represents the discrete set of measured time points. Note that for the PDE case we only consider the discrete time points where we have measurements, not the whole time domain. A well-trained model that adheres to the underlying physics should yield $\mu$ values approaching zero.

To assess the parameter estimation performance of the three techniques, we computed the relative root mean squared error between the true parameters and the estimated parameters:

$$\beta = \sqrt{\frac{1}{p} \sum_{j=1}^{p} \left( \frac{\eta_j^{\text{true}} - \eta_j^{\text{est}}}{\eta_j^{\text{true}}} \right)^2}$$

where $p$ denotes the total number of parameters in the differential equation.

Additionally, we evaluated the model performance by comparing the noisy observed data with the predictions obtained by solving the differential equations using the estimated parameters in absolute and relative terms:

$$\gamma_{\text{abs}} = \sqrt{\frac{1}{N_{\text{data}}} \sum_{j=1}^{N_{\text{data}}} \left( \hat{\boldsymbol{y}}_j - \boldsymbol{u}^{\text{pred}}(t_j, \boldsymbol{x}_j; \boldsymbol{\eta}^{\text{est}}) \right)^2}$$

$$\gamma_{\text{rel}} = \sqrt{\frac{1}{N_{\text{data}}} \sum_{j=1}^{N_{\text{data}}} \left( \frac{\hat{\boldsymbol{y}}_j - \boldsymbol{u}^{\text{pred}}(t_j, \boldsymbol{x}_j; \boldsymbol{\eta}^{\text{est}})}{\hat{\boldsymbol{y}}_j} \right)^2}$$

where $N_{\text{data}}$ represents the total number of data points, $\hat{\boldsymbol{y}}_j$ denotes the $j$-th noisy measurement vector, and $\boldsymbol{u}^{\text{pred}}(t_j, \boldsymbol{x}_j; \boldsymbol{\eta}^{\text{est}})$ is the predicted vector at the corresponding space-time point using the estimated parameters $\boldsymbol{\eta}^{\text{est}}$.

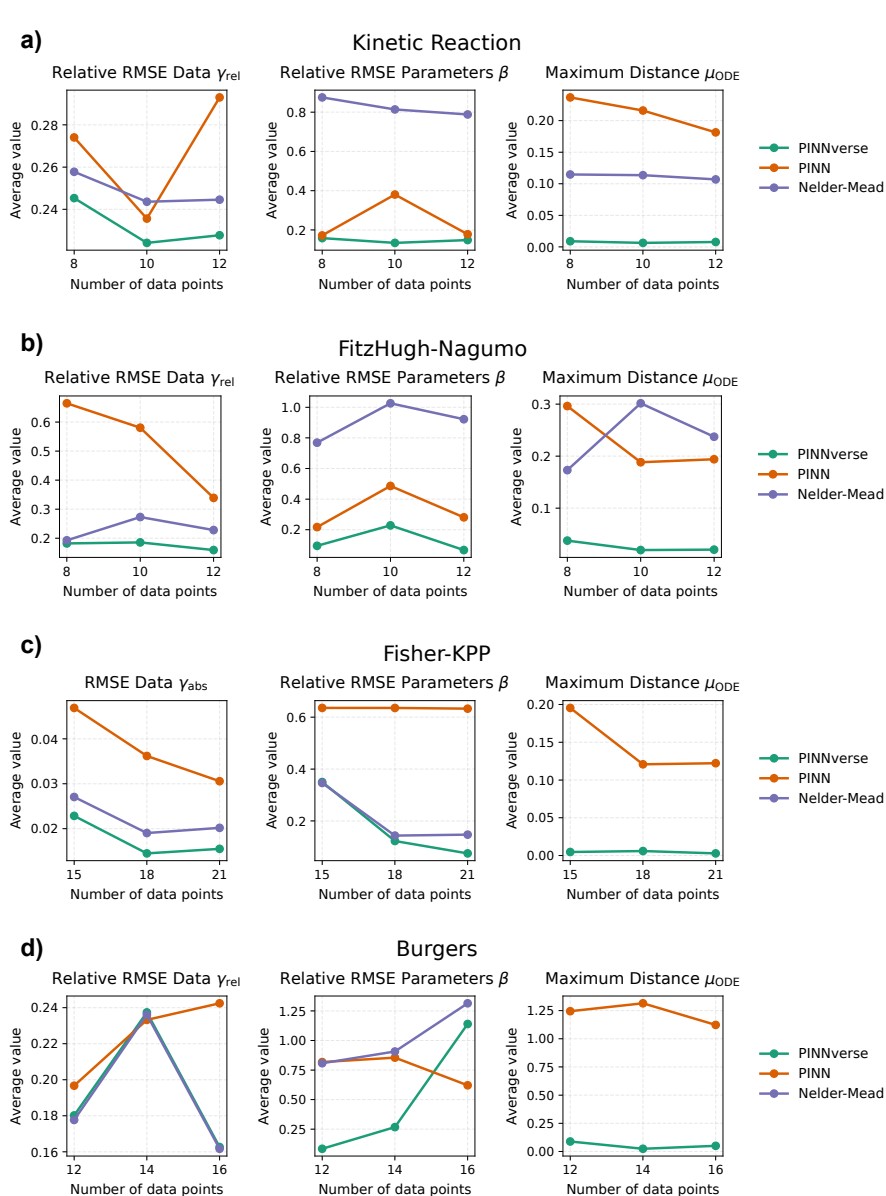

Figure 6: **Comparative performance of PINNverse, PINN, and Nelder–Mead methods across different numbers of noisy data points.** Each case was evaluated using standardized metrics: (i) Relative/Absolute RMSE Data Error ($\gamma_{rel}/\gamma_{abs}$), quantifying the deviation between predicted and observed data; (ii) Relative RMSE of Parameters ($\beta$), assessing accuracy of parameter estimation; and (iii) Maximum Distance ($\mu_{ODE}$), indicating the largest deviation from the true solution. PINNverse consistently outperforms the other methods across metrics, except for parameter estimation in the Burgers equation at 16 data points.

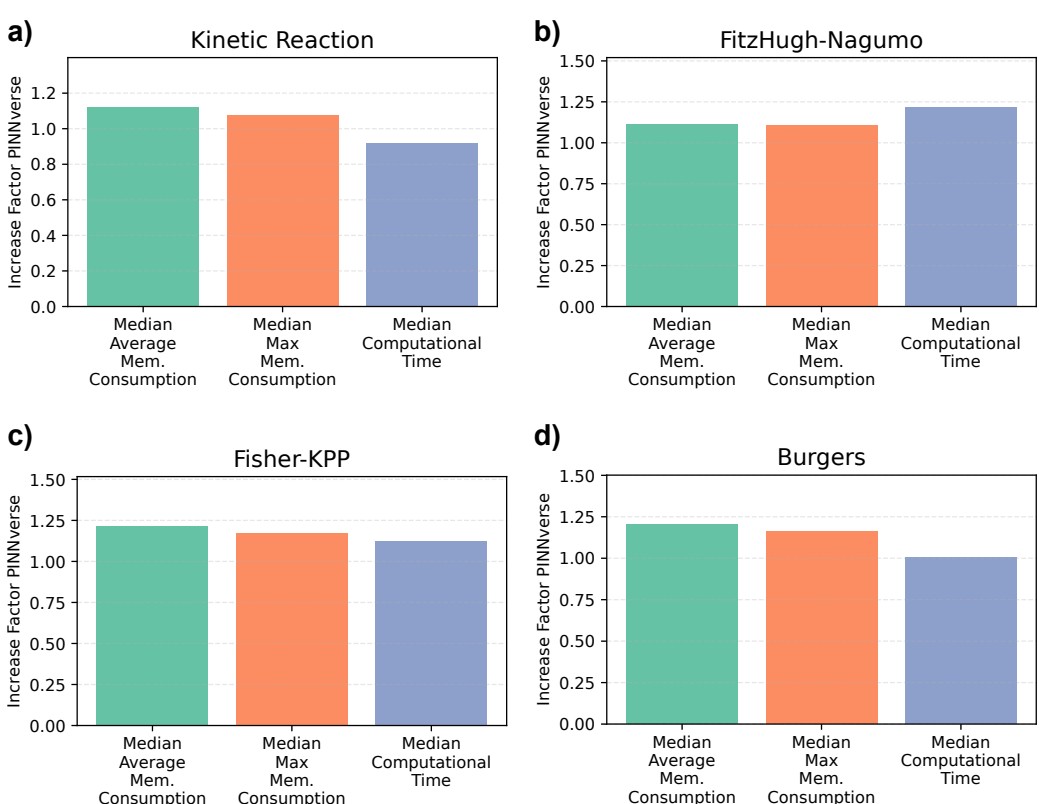

Figure 7: **Computational resource comparison between PINNverse and PINN.** Median values of the 'Increase Factor' (PINNverse relative to PINN) for memory consumption and computational time across four DE parameter estimation problems: Biodiesel (a), FitzHugh–Nagumo (b), Fisher–KPP (c), and Burgers (d). Results indicate a slight increase in computational time (1.0–1.5-fold) with PINNverse, while memory consumption remains nearly equal between the two methods across all test cases.

