# OpenReview forum: "PINNverse: Accurate parameter estimation in differential equations from noisy data with constrained physics-informed neural networks"
_ICLR.cc/2026/Conference — ICLR 2026 Conference Withdrawn Submission_

### Official Review · Reviewer_Bk6R · 2025-10-20

**Soundness:** 2
**Presentation:** 2
**Contribution:** 2
**Rating:** 2
**Confidence:** 3

**Summary:**

This paper tackles parameter estimation in differential equations from noisy observations. Instead of the standard weighted-sum objective used in PINNs, it formulates learning as constrained optimization: minimize data loss subject to physics (PDE/IC/BC) equality constraints. The method uses a Modified Differential Method of Multipliers (MDMM) and updates network weights, physical parameters, and Lagrange multipliers in a single loop. Across several benchmarks, it reports higher robustness and accuracy than vanilla PINNs and Nelder–Mead under noise and initial condition mismatch.

**Strengths:**

- (Novelty & Technical contribution) PINNverse successfully avoids nested optimization loops of traditional augmented Lagrangian methods, reducing computational costs. This delineates PINNverse from other models that use the augmented Lagrange method for PINNs.

- (Empirical contribution) PINNverse robustly infers parameters even with substantial data noise and poor initial guesses, scenarios where even robust classical numerical optimizers fail.

- (Broader impact) PINNverse is compatible with SOTA optimizers such as Adam.

**Weaknesses:**

- (Relevant citation) An important reference appears missing, which uses the augmented Lagrange method: “Enhanced Physics-Informed Neural Networks with Augmented Lagrangian Relaxation Method (AL-PINNs)“ [https://arxiv.org/abs/2205.01059](https://arxiv.org/abs/2205.01059). Could the authors discuss this work?

- (Presentation) My main concern is that the main text is not self-contained. The description of the proposed method, *PINNverse* (lines 162–196), relies heavily on Appendix A. Please refer also to the questions below.

- (Theoretical contribution) The paper would benefit from convergence analysis.

- (Empirical contribution) Hyperparameter ablations are recommended. Sweep penalty magnitude, dual LR, $\lambda$ initialization/schedule, and report sensitivity.

- (Reproducibility & Fair comparison) I could not find hyperparameter configurations for all models. Are all baselines properly tuned?

- (Reproducibility) Error bars are missing. Statistical analysis is important because PINNs tends to be sensitive to seeds and hyperparameters.

- (Reproducibility) It would be recommended to add README to the submitted code for interested readers.

### Minor Comment

- (Line 220) An upside-down exclamation mark should be corrected.

### Review Summary

Although the novelty and technical contributions are sound, the presentation requires improvement. In addition, I am concerned about the poor reproducibility and experimental settings of the results. Therefore, I am inclined to recommend rejection.

**Questions:**

- Why and how does PINNverse avoids nested loops? Could the authors elaborate on the following statement in the main text?:

  > (Lines 185-187) MDMM elegantly avoids this by updating the neural network parameters, differential equation parameters, and Lagrange multipliers all at once within a single backward pass.

---

> ### Author Response · Authors · 2025-11-20
>
> We thank the reviewer for the detailed feedback and for highlighting the strengths of our proposed approach.
>
> 1. We appreciate the pointer to AL-PINNs and will include a discussion of this work and its relation to ours in the revised manuscript.
> 2. We will correct the noted typo and do our best to improve presentation clarity by moving essential parts of Appendix A into the main text so that the description of PINNverse is self-contained. We would like to note that given that the reviewers generally requested extensions to the scope and presentation of the work, some compromises will be inevitable due to manuscript length limits. We hope to be able to keep these minimal.
> 3. To address concerns about reproducibility and empirical completeness, we hope to be able to add hyperparameter ablations covering penalty magnitudes and Lagrange multiplier initialization, and we are in the process of determining the sensitivity to random seeds. Please note that given the limited time available to comply with the different reviewer requests, we might only be able to analyze the sensitivity to random seeds.
> 4. We will document hyperparameter choices clearly in the revision and add error bars to reflect variability across runs from different random seeds. Note that we deliberately designed the numerical examples to have equal settings across all different methods to show that the advantage of PINNverse stems from its way of updating the parameters alone, which may be considered a form of ablation study itself.
> 5. We will include a README file to the code to improve clarity for readers, users and developers.
> 6. On the theoretical side, PINNverse inherits convergence guarantees directly from the MDMM method as established in Platt and Barr’s original analysis. We will clarify this connection in the revised text.
> 7. Finally, regarding the question about avoiding nested loops, we will expand our explanation: MDMM performs simultaneous updates of primal variables (neural network weights and PDE parameters) and dual variables (Lagrange multipliers) through a single backward pass, eliminating the inner optimization typically required in augmented Lagrangian frameworks. We will revise the relevant paragraph to make this core mechanism more apparent.
>
> We hope that we are able to address the reviewer’s points in a satisfactory manner in this way during the tight time limits of the reviewing period.

---

> ### Comment · Reviewer_Bk6R · 2025-11-26
>
> I appreciate the authors’ commitment to the discussion.
>
> Point 5 will be helpful for practitioners, and Point 7 addressed my question about avoiding nested loops.
>
> I assume the authors are currently revising the manuscript, so I would like to comment on my remaining concerns:
>
> - Could the authors elaborate on points 1, 3, 4, and 6?
>
> - I believe point 2 requires a major revision of the manuscript, which could be grounds for rejection.
>
> At this stage, I keep my current score.

---

### Official Review · Reviewer_2oPt · 2025-10-21

**Soundness:** 3
**Presentation:** 3
**Contribution:** 2
**Rating:** 4
**Confidence:** 4

**Summary:**

This paper proposes PINNverse, a novel training method that reframes the learning process of Physics-Informed Neural Networks (PINNs) as a constrained optimization problem, thereby further addressing the issue of loss weight balancing among multiple objectives in PINN training. The core of PINNverse is the Modified Differential Method of Multipliers (MDMM), which updates network weights and Lagrange multipliers (via gradient ascent) within a single optimization loop. Experiments on four differential equation benchmarks demonstrate that PINNverse achieves robust and accurate parameter estimation, even in the presence of significant data noise and poor initial guesses. It successfully prevents overfitting and ensures strict adherence to the governing physics.

**Strengths:**

1. The paper is well-written and easy to follow.
2. Experiments on multiple benchmarks show that PINNverse achieves better performance compared to vanilla PINNs.

**Weaknesses:**

1. Line 220: typo
2. Due to the highly non-convex nature of PINN-based differential equation solving, MDMM cannot guarantee strict satisfaction of hard constraints and PINNverse may still converge to local optima. Therefore, the authors' claim of "ensuring strict adherence to the governing physics" is unsubstantiated. The loss curves in Figures 2, 3, and 4 clearly show that the DE/IC/BC constraints are not strictly met during optimization. Experiments only confirm that PINNverse converges to superior solutions compared to the vanilla PINN.
3. Based on point 2, PINNverse more closely resembles a dynamic loss weighting adjustment method. Its weight update rule, based on MDMM, falls into the same category as existing adaptive weighting methods based on gradients or losses, differing primarily in the update formula and theoretical foundation. Intuitively, PINNverse not only introduces new loss terms ($L_i^2$ in Line 179) but also continuously increases the loss weights for PDE/IC/BC constraints during training (via the dual update in MDMM). These operations inevitably bias PINNverse towards reducing these constraint losses, making the model prioritize satisfying physical constraints. The reviewer recommends that it would be better to compare PINNverse with other SOTA dynamic loss weighting methods to clarify the advantages of introducing MDMM.
4. Subplots (c) in Figures 2, 3, 4, and 5 show a similar trend: PINN and PINNverse have comparable loss reduction in the early training stages. In the later stages, the data loss of the PINN continues to decrease, while the DE/IC/BC losses of PINNverse continue to decrease. The reviewer suspects this is related to a significant increase in the $\lambda_i$ during the training of PINNverse. Detailed epoch-$\lambda_i$ curves would help illustrate the influence of MDMM more clearly.
5. Based on point 4, the reviewer suggests to add an ablation study: train a vanilla PINN using the final weights for the DE/IC/BC losses obtained from PINNverse training. This ablation study would help clarify whether the performance improvement of PINNverse over the vanilla PINN is due to the larger physics term weights or the dynamic weight updates based on MDMM.

**Questions:**

Please refer to the weaknesses. If the authors can adequately address points 3 and 5, and convincingly demonstrate that employing MDMM for dynamic loss weighting is a superior approach, the reviewer would consider raising the score.

---

> ### Author Response · Authors · 2025-11-20
>
> We thank the reviewer for the constructive feedback and positive assessment of the clarity and experimental results.
>
> 1. We will correct the noted typo.
> 2. Regarding the claim of “ensuring strict adherence to the governing physics,” we agree that, due to the non-convex nature of PINN training, exact satisfaction cannot be guaranteed; we will relax the wording to indicate that the method enforces the constraints approximately/iteratively and typically to a significantly higher degree than standard PINNs.
> 3. To improve empirical completeness, we are in the process of including additional baselines, specifically SA-PINNs and h-PINNs, in the revision to more directly compare against other dynamic loss-weighting and hard-constraint methods.
> 4. Concerning the reviewer’s observation that the dual variables influence the prioritization of PDE/IC/BC losses, we are going to add plots showing the evolution of each Lagrange multiplier across epochs, allowing a clear visualization of how MDMM interacts with the loss landscape.
> 5. Following the reviewer’s suggestion, we will also conduct an ablation in which a vanilla PINN is trained using the final DE/IC/BC weights obtained from PINNverse; we expect this to clarify whether the improvements stem from dynamic updates rather than simply using larger fixed weights. These additions will make it clearer how MDMM differs from existing adaptive-weighting strategies and why its update mechanism yields more robust convergence.
>
> We appreciate the reviewer’s indication that addressing points 3 and 5 would meaningfully improve the assessment, and we will do our best to have these additional analyses included in the revised version.

---

### Official Review · Reviewer_gXZ1 · 2025-10-29

**Soundness:** 2
**Presentation:** 3
**Contribution:** 2
**Rating:** 6
**Confidence:** 5

**Summary:**

The paper introduces PINNverse, a reformulation of the standard Physics-Informed Neural Network (PINN) training process as a constrained optimization problem, solved using the Modified Differential Method of Multipliers (MDMM).  The paper directly targets a persistent weakness in the PINN framework that of unreliable parameter inference from noisy or sparse data, a very relevant problem.
Most PINN research focuses on soft-constraint or weighted-sum losses. Here physics residuals are used as hard equality constraints, allowing PINNverse to avoid the need for ad-hoc weight tuning and reach non-convex regions of the Pareto front, which standard PINNs cannot access.
While related to earlier hPINN/PECANN/ADMM-PINN work, the novelty lies in integrating MDMM with gradient-based deep learning optimizers (e.g., Adam) without nested loops, enabling efficient simultaneous updates of both primal (network/parameter) and dual (Lagrange multiplier) variables. The paper is largely complete but not exhaustive.  The methodology is sound and well-presented, but broader benchmarking and ablation studies would improve completeness.

**Strengths:**

The paper is well motivated, is clear to read and has a good set of relevant refernces.
Novelty: While related to earlier works, the novelty of this paper lies in integrating MDMM with gradient-based deep learning optimizers without nested loops, enabling efficient simultaneous updates of both primal (network/parameter) and dual (Lagrange multiplier) variables.
Relevance: Inverse problems for differential equations are ubiquitous in science and engineering. and very relevant especially for high dimensional complex systems such as battery chemistry models.
Results: Across 4 benchmarks (Kinetic Reaction, FitzHugh–Nagumo, Fisher–KPP, Burgers) are presented, however, results on real experimental datasets or high-dimensional PDEs woudl be more appreciated.

**Weaknesses:**

Missing the following:
1.Detailed ablation on penalty parameters, learning-rate effects, or sensitivity to multiplier initialization
2.Comparative results with adaptive weighting or multi-objective PINN variants (e.g., NSGA-PINN, hPINN), which would strengthen empirical completeness
3.Higher-dimensional PDEs or irregular geometries
discussions on the following would help
The convergence proof for MDMM in the stochastic, high-dimensional setting of neural networks is assumed but not formally analyzed.
No quantitative uncertainty or confidence interval analysis is presented for parameter estimates.
No results on real experimental datasets or high-dimensional PDEs.
Baseline PINNs use fixed unit weights.Stronger baselines (adaptive or Pareto-PINN) could better test robustness.
Scalability is not explored, only 1D/low-dimensional PDEs are tested; unclear behavior for 3D or coupled multiphysics systems.

**Questions:**

the approach depends heavily on simultaneous primal–dual updates. Given that deep-learning optimizers already operate on stochastic mini-batches, what guarantees do you have that the resulting stochastic saddle-point dynamics remain convergent or at least bounded?

PINNverse claims to access concave regions of the Pareto front. Could you visualize or quantify this property—perhaps by mapping Pareto trajectories or comparing multi-objective fronts empirically?  Figure(1) shown are schematics .

The paper assumes perfect PDE form and boundary conditions. How will the algorithm behave if there are modeling gaps or partially known boundaries—does the constraint formulation make the solution brittle?

In the Fisher–KPP and Burgers experiments, PINNverse’s advantage is pronounced under noise. Could this be partly due to the additional regularization implicit in the quadratic penalty term rather than the MDMM ?

---

> ### Author Response · Authors · 2025-11-20
>
> We thank the reviewer for the thorough and constructive assessment. We appreciate the recognition of the motivation, clarity, and relevance of the proposed MDMM-based formulation. Below we address the main concerns:
>
> 1. Regarding empirical completeness, we agree that additional ablations and baselines will improve the study. In the revision, we are working on including sensitivity analyses with respect to random seeds, multiplier initialization, and damping parameters.
> 2. We also agree that further comparisons to existing methods will be helpful, therefore we are in the process of adding comparisons with h-PINN and SA-PINN.
> 3. Furthermore we are also preparing an example with irregular geometry and an additional higher-dimensional PDE to broaden the scope of the evaluation.
> 4. Concerning theoretical aspects, MDMM’s convergence guarantees in the presence of stochastic gradient updates follow directly from the original analysis by Platt and Barr, where the method was specifically developed in the context of neural network training. We will clarify this connection more explicitly in the revised text.
> 5. With respect to uncertainty quantification, confidence intervals can be constructed post hoc when the data loss corresponds to a likelihood function, for example via profile likelihood or Wald-type approximations. In the revised manuscript, we will add a brief discussion on this.
> 6. Addressing the reviewer’s question on stochastic saddle-point dynamics, as noted above, we rely on the theoretical properties of MDMM established in its foundational work, and we will articulate this more clearly in the revised paper.
> 7. Regarding the Pareto front, computing the full front in our multi-objective setting (3–4 dimensions) would exceed the computational budget available during the rebuttal period and it would be difficult to visualize comprehensibly, which is why we resorted to a lower-dimensional schematic that illustrates the main aspect. However, the theoretical justification is well established: as shown in Ref. 78 by Das and Dennis (1997), weighted-sum methods cannot capture concave regions of Pareto fronts, while MDMM provides convergence to solutions satisfying the constrained formulation, which may lie in concave regions.
> 8. On modeling gaps or partially known boundaries, these can be incorporated as additional unknowns parameterized by auxiliary neural networks, and the MDMM formulation remains applicable. We will clarify this in the revised text.
> 9. Finally, concerning the noise experiments, the quadratic penalty is inherent to MDMM (removing it reduces the method to BDMM, which Platt and Barr showed to be inferior). The goal of MDMM is not to regularize the solution but to stabilize enforcement of equality constraints.
>
> Given the time constraints of the review period and manuscript space limitations, we might not be able to address all points completely by extending our work in all requested directions, as they entail both implementation and computational efforts. Our priority will be addressing points 2 and 3 first (which we deem the most valuable aspects to add), likely followed by point 1. We hope you understand this and consider our ongoing efforts sufficient for publication.

---

> > ### Comment · Reviewer_gXZ1 · 2025-11-26
> >
> > i believe the authors have made a good faith effort to clarify /address concerns within the constraints of time, my rating will remain the same.
> > thank you

---

### Official Review · Reviewer_aZ7S · 2025-10-31

**Soundness:** 2
**Presentation:** 3
**Contribution:** 2
**Rating:** 4
**Confidence:** 4

**Summary:**

This paper proposes converting PINN (a single-loop soft penalty method) into a hard-constrained optimization problem—treating data loss as the objective, with IC, BC, and PDE residuals as equality constraints, and bounded constraints on the PDE coefficient—using the Modified Differential Method of Multipliers (MDMM) to solve it, because PINN cannot reach concave parts of the Pareto front.

**Strengths:**

This paper clearly explains the problem PINN is facing and how converting to a hard constrained problem can help. Meanwhile, this paper clearly explains MDMM and its advantage.

**Weaknesses:**

1: This paper does not provide baseline methods for solving inverse problems. A well-known hard constrained paper is [1].

2: PINN is well-known for its inability to solve the forward problem when the PDE coefficient is large [2]. This paper does not demonstrate performance in predicting the inverse problem for high PDE coefficients.

3: This paper restricts PDEs to those with a single scalar coefficient. However, the PDE coefficient can also be a function. This paper needs additional experiments to demonstrate the proposed method's performance in such cases.

[1] Lu, Lu, et al. "Physics-informed neural networks with hard constraints for inverse design." SIAM Journal on Scientific Computing 43.6 (2021): B1105-B1132.

[2] Krishnapriyan, Aditi, et al. "Characterizing possible failure modes in physics-informed neural networks." Advances in neural information processing systems 34 (2021): 26548-26560.

**Questions:**

The method imposes a box (bound) constraint on the PDE coefficient, requiring it to lie between specified lower and upper limits. If the goal is to infer the coefficient, how are these bounds determined a priori? I’m not convinced that this constraint is justified.

---

> ### Author Response · Authors · 2025-11-20
>
> We thank the reviewer for the helpful comments and for noting the clarity of our motivation and MDMM formulation. We would like to address your concerns in the following way:
>
> 1. We agree that including additional baselines would strengthen the experimental section. In the revision, we plan to incorporate Hard-constrained PINNs (h-PINN) and Self-adaptive PINNs (SA-PINN). These baselines will be added to all inverse-problem benchmarks.
> 2. Furthermore, we appreciate the reviewer’s suggestion to evaluate cases known to be difficult for PINNs (Krishnapriyan et al., 2021). We are updating our Fisher–KPP experiment using the same high-contrast coefficient setup, enabling a direct comparison with the failure modes documented in the literature.
> 3. We also agree that extending beyond a scalar coefficient is valuable. The MDMM formulation naturally generalizes to functional coefficients parameterized by a neural network, and we are currently preparing an example where the unknown coefficient varies spatially.
> 4. Regarding the box constraints, the reviewer raises a valid point. In many physical or biological inverse problems, feasible parameter ranges are known from experimental considerations or physiological limits (e.g., diffusion coefficients or conductivities often lie in well-characterized ranges). For such problems, bound constraints help exclude physically implausible solutions. However, MDMM does not rely on box constraints; they remain entirely optional, and we can relax them to minimal assumptions such as ensuring positivity. The method continues to apply even without any bounds.
>
> We will try our best to address as many of the points during the limited remaining review period as possible. As the requested extensions require some implementation and simulation effort, we might not have working examples for all of them ready until the end of the review period. We will prioritize the points in the chronological order.

---

### Note · Authors · 2025-12-02

I have read and agree with the venue's withdrawal policy on behalf of myself and my co-authors.